# Wide-Pulse High-Frequency Neuromuscular Electrical Stimulation Evokes Greater Relative Force in Women Than in Men: A Pilot Study

**DOI:** 10.3390/sports10090134

**Published:** 2022-09-05

**Authors:** Xin Ye, Nathan Gockel, Daniel Vala, Teagan Devoe, Patrick Brodoff, Victor Gaza, Vinz Umali, Hayden Walker

**Affiliations:** Department of Rehabilitations Sciences, University of Hartford, West Hartford, CT 06117, USA

**Keywords:** isometric contraction, involuntary, sex-related difference, elbow flexor muscles, discomfort, sport rehabilitation

## Abstract

This study aimed to examine the potential sex differences in wide-pulse high-frequency neuromuscular electrical stimulation (WPHF NMES)-evoked force. Twenty-two subjects (10 women) completed this study. Prior to the stimulation, the visual analogue scale (VAS) for discomfort and the rating of perceived exertion (RPE) were measured, followed by the isometric strength of the dominant elbow flexor muscles. The subjects then completed ten, 10-s on 10-s off WPHF NMES (pulse width: 1 ms, frequency: 100 Hz) at maximum tolerable intensities. The subjects’ RPE was recorded after each set, and the VAS was measured following the last stimulation. The stimulation induced significant increase in discomfort for both sexes, with women having greater discomfort than men (men: 22.4 ± 14.9 mm, women: 39.7 ± 12.7 mm). The stimulation amplitude was significantly greater in men than in women (men: 16.2 ± 6.3 mA, women: 12.0 ± 4.5 mA). For the evoked force, only the relative NMES-evoked force was found greater in women than in men (men: 8.96 ± 6.51%, women: 17.08 ± 12.61%). In conclusion, even at the maximum tolerable intensity, WPHF NMES evoked larger relative elbow flexion force in women than in men, with women experiencing greater discomfort.

## 1. Introduction

Traditional resistance exercise training has been proven to be an effective tool to improve muscle strength, as well as to serve as a countermeasure to neuromuscular deterioration [1,2]. However, certain situations (e.g., injury, limb immobilization, prolonged bedrest) can hinder the applicability of such intervention. Thus, alternatives have been proposed and studied by researchers and clinicians, including neuromuscular electrical stimulation (NMES). NMES has been extensively used in clinical practice and sports medicine since late 1970s [3,4]. As an effective tool for some clinical populations going through rehabilitation [5], or an adjunct to voluntary resistance training for athletes [6], NMES can be beneficial for preserving muscle mass and neuromuscular functions. Besides, in an environment (e.g., microgravity) where the human body cannot lift weights, using NMES can also potentially be beneficial [7]. It has also been shown to be effective in treating muscle mass and strength decreases in aging populations [8,9]. Briefly, NMES involves delivering trains of electrical stimuli to superficial skeletal muscles or motor nerves via self-adhesive electrodes connected to a current generator. At relatively low force levels, the electrical stimuli can evoke strong involuntary muscle twitches by recruiting high-threshold motor units (thereby activating fast-twitch muscle fibers). Thus, the muscle activation pattern during the NMES is different from that during the voluntary contractions, where high-threshold motor units can only be recruited at high force levels.

A challenge that most practitioners face is the lack of detailed practice guidelines for selecting the appropriate and effective electrical current pulse parameters, such as the pulse duration/width and frequency. The relatively short pulse width (50–400 µs) and low frequency (15–40 Hz) NMES has been used in many NMES interventions studies. This type of NMES mainly activates motor axons with little influence on the sensory fibers [10]. However, by increasing both the pulse width (e.g., 1 ms) and frequency (e.g., 100 Hz), significantly larger stimulation-evoked involuntary muscle contractions can be achieved [10] (sometimes referred to as “extra torque”), due to the additional recruitment of spinal motoneurons by the electrically evoked sensory fibers [11,12]. A recent review article suggested using wide-pulse, high-frequency (WPHF) NMES for evoking a greater force and better adaptations to the NMES training [13].

An important fact of applying WPHF NMES on human muscles that needs to be mentioned is the large variability of the evoked muscle force levels, which makes standardizing NMES current pulse parameters even harder. For example, studies [14,15,16] have consistently reported so-called “responders” (the proportion of subjects showing an extra torque during WPHF NMES) and “non-responders” to the WPHF NMES protocols. Many factors can influence the WPHF NMES-evoked force, but interestingly, the sex-related factor has not been previously examined and well-documented. Part of the reason may be due to the relatively smaller female sample size in many of the previous studies. Considering the sex difference in proportional area of type I fibers in the skeletal muscle (e.g., women overall have greater relative area for type I fiber, see a review in Hunter (2016) [17], but men have significantly larger type I fiber areas and mean fiber areas (than the women in biceps brachii [18]), it is therefore possible that this factor can influence the WPHF NMES-evoked force. Theoretically speaking, activating a muscle with greater proportion of type I fiber would generate less force than activating a muscle with greater proportion of type II fiber.

Therefore, the purpose of the study was to explore if there is a sex-difference in WPHF NMES-evoked force in elbow flexor muscles under maximum tolerable stimulation intensities. Additionally, we also aimed to examine if the WPHF NMES-induced discomfort may differ between sexes. The results of the study can potentially provide useful and important information for clinicians and trainers, as currently there is no specific guideline suggesting the proper use of WPHF NMES in both sexes.

## 2. Materials and Methods

### 2.1. Study Design

The main purpose of this study was to examine the potential sex-related differences in WPHF NMES-evoked involuntary force as well as discomfort level on the upper limb elbow flexor muscles. A familiarization session in the laboratory was scheduled and conducted prior to the experimental visit, during which the subjects were given the opportunity and time to familiarize with all the experimental and testing procedures. Additionally, the WPHF NMES amplitude was determined during the familiarization. At least one week after the familiarization session, the subjects returned to the laboratory, when the experiment was conducted.

### 2.2. Subjects

Twelve men (age: 23 ± 6 years; height: 179 ± 7 cm; weight: 86 ± 15 kg) and ten women (age: 23 ± 3 years; height: 162 ± 8 cm; weight: 59 ± 6 kg) voluntarily participated in and completed this study. All subjects were healthy and were free from any current or recent (1 year) injuries, neuromuscular, or musculoskeletal disorders in the upper body parts, especially the shoulders, elbows, wrists, and fingers. Prior to any experimental testing or procedures, all subjects signed an informed consent and completed a pre-exercise health and exercise status questionnaire. During the consent process, the subjects were instructed to maintain their normal daily activities regarding dietary intake, hydration status, and sleep. Additionally, they were not allowed to perform any vigorous physical activities and resistance exercises during the entire study. This study was approved by the University Institutional Review Board (Protocol ID#: PRO02021000209), and was conducted in conformity with the policy statement regarding the use of human subjects by the Declaration of Helsinki.

### 2.3. Procedures

#### 2.3.1. Familiarization

The first visit of the experiment was to familiarize the subjects with the testing measurements as well as the WPHF NMES. Upon arrival, the subjects’ standing height and body weight were taken. They were also asked which hand they would prefer to throw a ball or perform a heavy punch, to determine the upper limb dominance. This followed by the familiarizations of the rating of perceived exertion (RPE) scale for how hard the subjects felt their muscle was working, along with the visual analogue scale (VAS) for discomfort. Then the subjects were familiarized with the elbow flexion isometric contraction, which was performed with the upright sitting position in front of a custom-built testing table and apparatus. Based on the limb dominance, the sitting position and the isometric contraction testing apparatus were adjusted. Specifically, the subject’s dominant side of the armpit was pushing firmly against a foam pad on the testing table, so the dominant arm was resting comfortably on the pad. At the same time, the subject’s dominant writ was attached by a strap, which was connecting to a load cell (Model SSM-AJ-500; Interface, Scottsdale, AZ, USA). The other end of the load cell was connected to an immovable frame, so an isometric elbow flexion muscle action could be performed. The chair height and the link length between the strap and load cell were adjusted for every subject to ensure the upper arm was parallel to the floor, and the elbow joint was at a 90-degree angle when performing an isometric elbow flexion contraction. With this setup, the subjects practiced contracting their dominant elbow flexor muscles against the strength testing apparatus a few times with about 50% of the maximal effort, then followed by two to three times of the maximal effort.

The last portion of this visit was to familiarize with the WPHF NMES. The subject’s dominant biceps brachii muscle belly was cleaned with rubbing alcohol, and shaved with a razor to remove surface hair. Then the investigators placed two stimulating electrodes (2 × 2-inch square TENS Unit Pads, AUVON Inc., Peachtree Corners, GA, USA) were placed on the proximal belly (cathode) and the distal tendon (anode) of the biceps brachii muscle [19]. A high-voltage (maximal voltage 400 V) constant-current stimulator (DS7R; Digitimer, Hertfordshire, England, UK) along with a trigger signal generator (High Precision DDS Signal Generator Counter, Koolertron, Hong Kong, China) were used to deliver the WPHF NMES (1 ms, 100 Hz, biphasic square waveform). Starting from zero amplitude, the investigator slowly and gradually (roughly 1 mA per second for most subjects) turned up the amplitude knob of the DS7R stimulator, and the subjects were asked to relax as much as he/she could. The subjects then informed the investigator to stop once they felt strong discomfort or pain. At least three 10-s WPHF NMES trials were performed for each subject, and the highest amplitude the subject achieved was recorded as the maximum tolerable amplitude. Lastly, the subjects were asked to point a number on the 6–20 Borg scale, and to make a mark on the VAS scale.

#### 2.3.2. Experimental Visit

At least seven days after the familiarization, the subjects returned to the lab for the experimental visit. The RPE and VAS were taken upon the subjects’ arrival as the baseline to ensure the subjects did not have physical exertion and discomfort. This was followed by the isometric strength testing of the dominant elbow flexors, during which the same manner was used as during the familiarization. Specifically, several submaximal elbow flexion isometric muscle actions were performed as the warm-up, followed by three 5-s maximal voluntary isometric contractions (MVICs), with 1 min between consecutive trials. At least 10 min after the isometric strength testing, the subjects went through 10 sets of 10-s on 10-s off WPHF NMES. The main reason we chose 10 sets is that this number is similar to the number of sets most training or rehabilitative programs utilize. Before the first WPHF NMES set, the subjects were re-familiarized with the WPHF NMES by getting stimulation with the highest amplitude achieved during the familiarization visit. All subjects in this study were able to tolerate this amplitude. Thus, the first stimulation set used the highest amplitude used in the familiarization visit. Following each stimulation set, the investigators asked the subjects to point the RPE during the set, and if they could try a higher stimulation amplitude for the next set. If so, the investigator would increase the stimulation amplitude by 1 mA for the following stimulation set. Once all 10 sets of WPHF NMES were completed, the subjects were asked to make a mark on the VAS to rate the discomfort for the entire stimulation sets. The experiment then was concluded.

### 2.4. Measurements

#### 2.4.1. Isometric Strength and WPHF NMES-Evoked Force

The force signals from the load cell during all the isometric strength testing and the WPHF NMES were collected via a load cell adapter (Trigno Load Cell Adapter, Delsys Inc., Natick, MA, USA), sampled at 2222 Hz with a wireless system (NeuroMap System, Delsys Inc., Natick, MA, USA), and then stored in a laboratory computer for further analyses. For each 5-s MVIC, the peak 1 s window was chosen and then calculated as the maximal force output. The average of the three maximal force outputs was then calculated as the maximal isometric strength.

For the WPHF NMES-evoked force, the entire 10-s stimulation period for each set was selected, and the absolute evoked force was calculated by averaging the force data during the 10-s stimulation period. This number was then used to divide by the maximal isometric strength, to calculate the relative evoked force level (%MVIC).

#### 2.4.2. RPE

A 6–20 Borg RPE scale [20] was used in this study. Even though the subjects did not voluntarily contract the elbow flexor muscles to induce physical exertion, the WPHF NMES was used to induce involuntary muscle contractions. To avoid any confusion, the investigators specifically told the subjects: “Imagine seated resting as a 6-No exertion at all, and performing a maximal muscle contraction such as our elbow flexion maximal isometric contraction as a 20, how hard your muscle worked from getting the electrical stimulation?”

#### 2.4.3. Discomfort–VAS

The discomfort was assessed using a 100-mm VAS. The scale shows “no discomfort at all” on the far-left side and “Unbearable discomfort or pain” on the far-right side. The subjects were asked to mark a vertical line on the VAS scale representing the discomfort level for the entire 10 sets of WPHF NMES.

### 2.5. Statistical Analyses

All results were presented as mean ± standard deviation (SD). The Shapiro-Wilks test was used to check and confirm that the dependent variables were normally distributed. The independent-samples t-test was used to examine the elbow flexion isometric strength between sexes. Separate two-way mixed factorial analysis of variance (ANOVA) tests were used to examine the sex-related differences in RPE [time (baseline, set 1, set 2, … set 10) × sex (men, women)] and discomfort [time (baseline, post-stimulation) × sex (men, women)]. Additionally, WPHF NMES amplitude, absolute evoked force, and relative evoked force were also examined using two-way mixed factorial ANOVA tests [time (set 1, set 2, … set 10) × sex (men, women)]. When appropriate, the follow-up tests included one-way repeated-measures ANOVA with Bonferroni-adjusted pairwise comparisons, as well as independent t-tests. The partial η^2^ statistic was provided for all the repeated measure comparisons, with values of 0.01, 0.06, and 0.14 representing small, medium, and large effect sizes, respectively [21]. All the statistical tests were conducted using statistical software (IBM SPSS Statistics 26.0; IBM, Armonk, NY, USA) with an alpha set at 0.05.

## 3. Results

There was a statistically significant difference for the elbow flexion isometric strength between sexes, with men showing 303.48 ± 71.52 N, and women 158.98 ± 27.57 N (*p* < 0.001). For the RPE, the two-way ANOVA did not show a significant time × sex interaction (*p* = 0.957) or main effect for sex (*p* = 0.255), but there was a significant main effect for time (*p* < 0.001). Figure 1 showed the changes in the RPE throughout the experiment. The two-way ANOVA revealed a significant time × sex interaction (*p* = 0.006) for the VAS. Thus, the follow-up independent *t*-tests indicated a significant difference at the post-stimulation time point between sexes (men: 22.4 ± 14.9 mm, women: 39.7 ± 12.7 mm, *p* = 0.009). Additionally, the discomfort level for both men (baseline: 0.2 ± 0.7 mm, post-stimulation: 22.4 ± 14.9 mm, *p* < 0.001) and women (baseline: 1.5 ± 2.6 mm, post-stimulation: 39.7 ± 12.7 mm, *p* < 0.001) significantly increased following the WPHF NMES sets (Figure 2).

The two-way ANOVA for the WPHF NMES amplitude did not show a time × sex interaction (*p* = 0.834), however, there were significant main effects for time (*p* = 0.024) and sex (*p* = 0.047). After collapsing across time, the pairwise comparison showed significantly greater WPHF NMES amplitude in men than in women (men: 16.2 ± 6.3 mA, women: 12.0 ± 4.5 mA, *p* = 0.046) (Figure 3a). There was no significant time × sex interaction or main effects for the absolute stimulation-evoked force (Figure 3b). For the relative evoked force, the two-way ANOVA did not show a time × sex interaction (*p* = 0.539) nor the main effect for time (*p* = 0.647), however, there was a significant main effect for sex (*p* = 0.041). After collapsing across time, the pairwise comparison showed significantly greater relative evoked force in women than in men (men: 8.96 ± 6.51%, women: 17.08 ± 12.61%, *p* = 0.033) (Figure 3c).

## 4. Discussion

The main purpose of this study was to investigate the potential sex-related difference in WPHF NMES-evoked involuntary muscle force. The main findings of the study are: (1) Maximum tolerable WPHF NMES induced greater discomfort in women than in men; (2) The amplitude of the maximum tolerable electrical stimulation was overall greater in men than in women; and (3) Even though there was no difference in absolute stimulation-evoked force, women experienced a greater relative evoked force than men did.

A number of previous studies [14,15,16,22,23,24,25,26] have examined WPHF NMES on force generation level and neuromuscular fatigue. But to our knowledge, this is the first study to examine the potential sex difference in an unconventional electrical stimulation WPHF NMES-evoked force. It is also important to mention, that different from most of the previous research where the stimulation current intensity was set to evoke a certain force level (e.g., 5% or 10% of the MVIC), the current investigation instead directly used the subjective rating of discomfort (VAS). Indeed, controlling the stimulation current intensity based on the same force level allows better comparisons among different conditions such as WPHF NMES, conventional NMES, voluntary muscle contractions, and so on. However, the discomfort-based stimulation intensity determination may be more practically meaningful, as practitioners such as therapists and trainers usually set the stimulation intensity based on the individuals’ discomfort level.

The first interesting finding of this study is the sex difference in the discomfort level after the WPHF NMES sets: even though the stimulation current intensity was set at maximum tolerable level in every stimulation set for all the subjects, women experienced greater overall discomfort than men did throughout the WPHF NMES. This is different from what we expected. Earlier experiments [27] as well as recent reviews [28,29] suggest that in general, men have higher pain tolerance and lower pain sensitivity than women. This was demonstrated in the current study from the results of the WPHF NMES current intensity: men had a significantly greater stimulation amplitude than women (4 mA difference in average). Thus, with the lower stimulation current intensity throughout the experiment, it is interesting to notice that the post-stimulation rating of discomfort (VAS) was greater in women than in men. A possible explanation may be provided by the data of the RPE throughout the WPHF NMES. Even though we did not find the significant sex × time interaction or main effects, the effect size of sex factor partial η^2^ = 0.064, indicating a medium effect size. As shown in Figure 1, therefore, the stimulation tended to impose a medium effect to induce a higher RPE in women than in men. After 10 sets of WPHF NMES, women might have felt that their muscles were more worked or sore than men did, which could have contributed to the greater discomfort scale. The sex-difference in discomfort is also likely to be explained by the different relative WPHF NMES-evoked force (discussed below), with a higher muscle contraction intensity, it is not surprise to see women had greater discomfort than men did.

The most interesting finding of the experiment is perhaps the sex-related difference in WPHF NMES-evoked force. There was no significant sex difference in the absolute evoked force, even though men had a greater stimulation intensity. Interestingly, when examining the effect size of the sex factor for the absolute evoked force, partial η^2^ = 0.053 belongs to the effect size range between small and medium. It can be seen from Figure 3b, that the absolute evoked force tends to be greater in women than in men, especially during the later WPHF NMES sets. Given that the elbow flexion isometric strength was nearly twice in men as comparing to women in the current sample, it is not surprising to see significant sex difference for the relative evoked force (average relative force men vs. women: 9% vs. 17% MVIC). Beside stimulation intensity, several other factors such as muscle size, subcutaneous fat thickness, and muscle fiber type could influence the electrically evoked muscle force level. For example, men in general have larger muscle cross-sectional area (CSA) and muscle fiber area for the biceps brachii than women [18]. If this were the case in the current study, we can speculate that using the same size of stimulating electrodes on women may activate a relatively larger area of muscle fibers than on men, which could be the reason for the greater relative stimulation-evoked force in women. Unfortunately, muscle size, along with other important factors, were not quantified in the current study, which made it difficult to further explore the underlying mechanisms for the sex-related difference. It is worth investigating in the future for the potential influences of these factors on the NMES-evoked force and discomfort.

Even though we had a few interesting findings, this experiment does have its own limitations. First, we only focused on the WPHF NMES for the current investigation, but did not examine if there are any sex differences for the conventional NMES. This is mainly due to the very low conventional NMES-evoked force level (<5% MVIC for most subjects, also requiring a much larger stimulation intensity) from our pilot testing. Second, a larger sample size including populations other than healthy adults can provide more useful information. The current study only examined normal healthy adults, and it’s important to know that the current findings may not be true in another population, such as clinical or athletic populations, who may benefit more from utilizing NMES more for various purposes. Third, as mentioned earlier, some potential influencing factors should have been examined to explore potential mechanisms accounting for the sex differences.

## 5. Conclusions

In conclusion, a bout of maximum tolerable intensity WPHF NMES (ten, 10-s on and 10-s off) induced greater overall discomfort in women than in men, even with higher stimulation intensity used in men. Sex difference in the absolute stimulation-evoked elbow flexion force was absent, but the relative evoked force was almost twice in women as compared to that in men. Even though this initial exploratory work was not able to identify any factors or mechanisms causing the observed sex differences, it can provide important practical information. Clinicians should keep this in mind when prescribing NMES to patients or clients, as women may seem to be more responsive to this treatment/intervention. Future work should focus on identifying influential factors and even possibly developing predictive equations, so practitioners can use such information to better quantify the NMES treatment/intervention.

## Figures and Tables

**Figure 1 sports-10-00134-f001:**
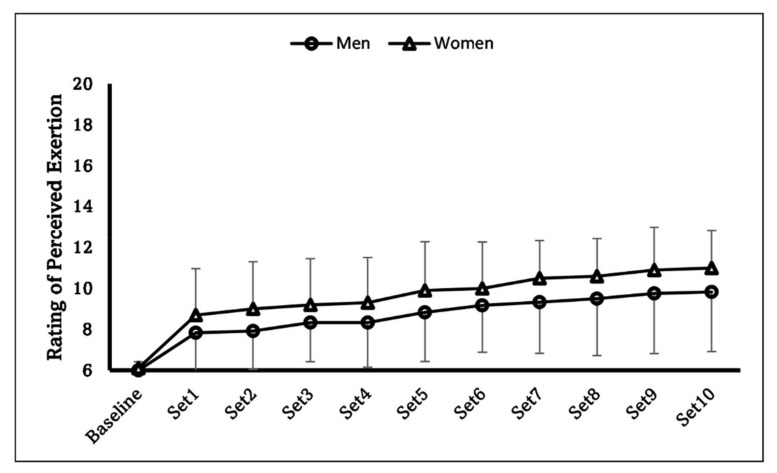
Changes of the RPE throughout the experiment for both sexes. Significant main effect for time was found.

**Figure 2 sports-10-00134-f002:**
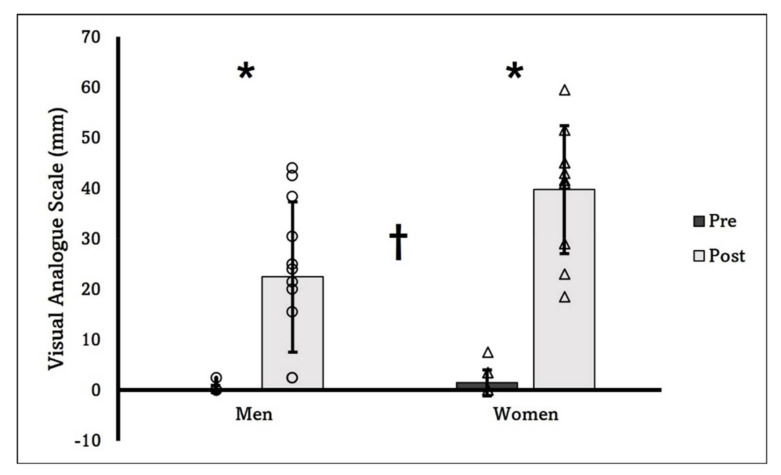
Changes of the VAS from pre- to post-stimulation for both sexes. Individual dots (circle: men; triangle: women) represent individual data. Significant sex × time interaction was found, with both sexes having higher VAS at the post time point (*****). Additionally, a significant great value at the post time point was found in women than in men (†).

**Figure 3 sports-10-00134-f003:**
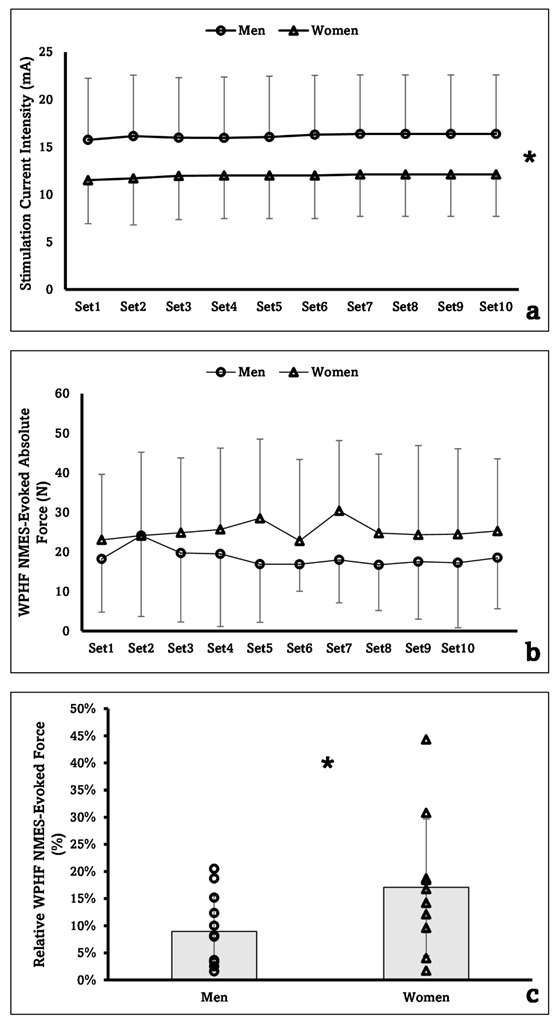
Changes of the stimulation current intensity (**a**), wide-pulse high-frequency (WPHF) NMES-evoked absolute force (**b**), and WPHF NMES-evoked relative force (**c**) for both sexes during the stimulation sets. Individual dots (circle: men; triangle: women) represent individual data. ***** Indicates significant main effect for sex.

## Data Availability

The data presented in this study are available on request from the corresponding author. The data are not publicly available due to the ongoing investigation not being completed.

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
