# Peer review of "Wide-Pulse High-Frequency Neuromuscular Electrical Stimulation Evokes Greater Relative Force in Women Than in Men: A Pilot Study"

_sports, 2022, doi:10.3390/sports10090134_

Round 1

Reviewer 1 Report

TITLE:

Given the small size of the sample, I suggest adding the phrase "a pilot study" to the title.

INTRODUCTION:

NMES frequency of 15-40 Hz is called ‘traditional’ in the manuscript. Should therefore a 60-110 Hz frequency be understood as ‘non-traditional’ even though it has long been used to stimulate fast-contracting muscle fibers (type II)?

The sentence “This type of NMES favors the activation of motor axons rather than the sensory axons [10].” seems to contain an undertone of criticism– but then the purpose of muscle stimulation is to activate the motor fibers.

SUBJECTS

This study does not provide a sample size calculation.

It does not follow the CONSORT guidelines for clinical trials.

PROCEDURES

Why was the anode placed on the tendon that does not contract rather than at the end of the muscle belly?

According to the description provided, stimulation started with a unipolar (monophasic) current. Why was the cathode positioned proximally when placing it at the distal end of the muscle would generate stronger contractions?

STATISTICAL ANALYSES

Did all variables have normal distributions when the sample size was so small?

DISCUSSION

How can physiotherapists benefit from the information that electrical stimulation causes more discomfort to men than to women when maximum muscle contractions are always induced by a current of maximum intensity tolerable by the patient?

Author Response

Thank you for the comments. Please see our point-by-point response in the attached document.

Reviewer 2 Report

Why was the VAS not measured at similar timepoints than the RPE? The rationale needs to be provided and discussed.

Would a comparison between male and female not only be interesting when they produce similar percentage of their MVC force with the electrical stimulation. It seems the %force was higher for females so no surprise then that the VAS scores were higher for females. This needs to be discussed.

In the abstract, the aim is not clear. It seems the interest in the responses when participants getting maximum tolerable intensity. Should that not be in the aim? Please reconsider.

L66. Have females a different fibre type distribution in elbow flexors. Would be great to provide that information.

L83. Change “all” to “with all”.

L84. Why was it an attempt and not a determination or measurement? Please clarify as it reads as if it was not always successful.

Ls 88-89. Do we the accuracy for height and body mass? I suggest to express without decimal places.

L105. Would throw a ball not being enough to determine limb dominance?

L107. “rating of perceived exercise”. This is an incorrect description of the RPE. Please change.

L107. Please be more specific as “physical exertion” is too general.

L108. Change “were then” to “were”.

L124. How do you know that dead skin was removed with razor. Normally sand paper is used for that. Please clarify.

L137. 10-point of 15-point Borg scale.

L142. The RPE was taken on arrival for physical exertion. Are you sure this was done?

L152. How could subjects achieve an amplitude. Please revise/clarify.

L160. Measurements.

L175. “the WPHF NMES could cause the muscles to contract involuntarily”. That is confusing, as the stimulation is used induce involuntarily contractions.

Figure 1. The scale of the y-axis needs to be changed as the RPE values can never be lower than 6.

Author Response

(The authors gave the same response as above.)

Round 2

Reviewer 2 Report

Thank you for addressing my comments and suggestions.